# Circadian Disruption and Sleep Disorders in Astronauts: A Review of Multi-Disciplinary Interventions for Long-Duration Space Missions

**DOI:** 10.3390/ijms26115179

**Published:** 2025-05-28

**Authors:** Hongjie Zong, Yifei Fei, Ningang Liu

**Affiliations:** State Key Laboratory of Radiation Medicine and Protection, School of Radiation Medicine and Protection, Collaborative Innovation Center of Radiological Medicine of Jiangsu Higher Education Institutions, Soochow University, Suzhou 215123, China; 18036559000@189.cn (H.Z.); feiyifeidoct@163.com (Y.F.)

**Keywords:** human spaceflight, circadian rhythm disruption, sleep optimization, spaceflight medication, countermeasure

## Abstract

As humanity advances into deep space exploration, astronauts on long-duration missions face significant challenges posed by circadian rhythm disruptions and sleep disorders, which arise from extreme environmental stressors such as microgravity, ionizing radiation, and operational workload. These disruptions not only compromise physiological and psychological health but also impair cognitive function and mission-critical performance. In this review, we summarized established countermeasures encompassing pharmacological interventions, light-based circadian regulation, and work–rest schedule optimization alongside innovative approaches such as gut microbiota modulation and traditional Chinese medicine.

## 1. Introduction

The circadian rhythm is an internal timing mechanism that organisms have evolved to adapt to the Earth’s 24 h rotation, important for maintaining physiological health, cognitive function, and work performance [1]. Human sleep–wake cycles represent a core component of this regulatory system, governed by periodic fluctuations aligned with diurnal light–dark cycles. As global space exploration progresses toward long-duration missions, astronauts encounter extreme environments lacking the natural 24 h light–dark periodicity essential for circadian stability. Maintaining astronaut health (both physical and psychological) and operational efficiency thus emerges as a critical challenge for deep space missions [2].

Numerous environmental stressors in spaceflight, including altered lighting conditions, microgravity, noise, cognitive workload, confinement, and space radiation, disrupt circadian rhythms and sleep architecture [3,4,5,6]. Initial observations from Skylab missions (1976) first documented alterations in rapid eye movement (REM) sleep [7], later corroborated by Mir station studies in the 1990s, which reported circadian desynchronization among astronauts [8]. While adequate sleep and alertness maintenance are essential for executing mission-critical tasks, increasing evidence indicates that circadian misalignment and sleep dysfunction are common among astronauts. The combined effects of sleep deprivation, circadian disruption, prolonged wakefulness, and high workload elevate the risk of operational errors, a paramount safety concern for current and future space missions [9]. Notably, NASA classifies sleep deficiency and circadian rhythm disorders as Category 1 risks for long-duration spaceflight [3], underscoring the urgent need for strategies to stabilize circadian function and optimize sleep for crew health and mission safety.

Countermeasures for astronauts circadian and sleep disorders represent a frontier in aerospace medication [3,10,11,12]. Existing approaches, including pharmacological interventions, spectral lighting regimens, and cognitive–behavioral approaches exhibit variable efficacy and implementation challenges in space environments. Meanwhile, emerging biotechnological approaches including gut–brain axis modulation and phytotherapeutic interventions show promise for translating terrestrial research into space-adaptive solutions. This study reviews advancements in countermeasures and therapeutic interventions for astronaut sleep deprivation and circadian rhythm disruptions, synthesizing evidence-based references for sleep medication research in long-duration manned missions.

## 2. Pharmacological Interventions

Pharmacological interventions constitute the primary therapeutic modality for managing sleep disturbances during spaceflight missions. Analysis of 79 U.S. spaceflight missions by Putcha et al. demonstrated sleep medications accounted for 45% of all pharmacological interventions [13]. Between 2001 and 2011, 78% (61 of 78) of shuttle crewmembers reported using sleep medications during space missions [14], a trend corroborated by survey data showing 71% of respondents relied on pharmacotherapy to initiate sleep [3]. Current medications in the U.S. astronaut formulary include Zolpidem, Zaleplon, Melatonin, Temazepam, Eszopiclone, and Quetiapine fumarate, while Chinese space protocols additionally incorporate Triazolam and Diphenhydramine, the latter offering dual benefits for motion sickness management [2].

Despite their widespread use, sleep medications often exhibit suboptimal efficacy for astronaut insomnia. Putcha et al. reported anecdotal evidence of reduced efficacy during space shuttle flights, identifying 13 different drugs, including Zolpidem, Quetiapine fumarate, and Temazepam, as “ineffective” or “mildly effective” for treating symptoms [13]. For example, Zolpidem showed only 1.7% “mildly effective” (1 of 58 administrations) and 7% “ineffective” (4 of 58) outcomes, Quetiapine fumarate had 6.8% “mildly effective” (3 of 44) and 6.8% “ineffective” (3 of 44), and Temazepam recorded 1.6% “mildly effective” (6 of 387) and 1.8% “ineffective” (7 of 387) results. Overdosing or polypharmacy occurred in approximately 17% of sleep medication nights (87/500) [3]. Dose control is critical for mission safety: a double-blind, placebo-controlled trial in astronauts revealed that low-dose Zolpidem (5 mg) and Zaleplon (10 mg) minimally impacted cognitive performance, whereas high-dose Zolpidem (10 mg) caused significant post-awakening cognitive impairment [15]. These findings highlight inconsistent efficacy in space environments, along with risks of dependency or inappropriate multi-drug use, necessitating deeper analysis of contributing factors to safeguard crew health and mission success.

Terrestrial drug efficacy profiles frequently fail to translate to space due to altered physiological responses, underscoring the need for space-specific pharmacodynamic models. Spaceflight stressors, such as microgravity, space radiation, hypercapnia, noise, and psychosocial stress, disrupt drug absorption, distribution, metabolism, and excretion (ADME) processes through multiple physiological pathways. For instance, simulated microgravity reduces hippocampal γ-aminobutyric acid (GABA), a key inhibitory neurotransmitter, while increasing excitatory glutamate in rats, disrupting the GABA-glutamate balance [16,17]. Zolpidem and similar sleep medications enhance GABAA receptor sensitivity to potentiate chloride influx and neuronal inhibition, but space-induced GABA depletion limits this mechanism, as insufficient neurotransmitter availability hinders inhibitory signaling despite receptor sensitization. Concurrently, microgravity-induced glutamatergic hyperactivity may further antagonize GABAergic inhibition. This dual dysregulation—weakened GABAergic inhibition and heightened glutamatergic excitation—compromises Zolpidem’s GABAA-dependent mechanism, reducing its sedative effects in space. In future spaceflight missions, the development of multi-target agents, such as those combining GABA agonists with glutamatergic signaling inhibitors, may reduce the required dosage of individual agents and mitigate side-effects through synergistic effects, thereby improving astronauts’ sleep quality.

The space environment can also significantly influence the activity of hepatic drug-metabolizing enzymes. Studies demonstrate that following a 30-day spaceflight, murine hepatic levels of CYP2C29, CYP1A2, and CYP2E1 exhibited 1.4-, 1.9-, and 1.8-fold increases, respectively, with CYP2E1 maintaining elevated expression during the recovery phase [18]. While sleep medications (e.g., Zolpidem, Eszopiclone) predominantly undergo CYP3A4-mediated metabolism, cross-talk between CYP isoforms may indirectly modulate their metabolic clearance rates, thereby potentially altering pharmacological efficacy. Existing studies document reduced drug absorption rates with significantly delayed pharmacokinetic profiles during spaceflight, which may contribute to Zolpidem overuse as a potential contributing factor. Acetaminophen serves as the gold standard probe drug for assessing absorption pharmacokinetics. In the early stages of flight, acetaminophen pharmacokinetics in astronauts demonstrated reduced Cmax and prolonged Tmax. Subsequent cohort studies confirmed these absorption impairment patterns; International Space Station (ISS) trials (=5) revealed extended Tmax but preserved AUC (area under the curve) for oral acetaminophen formulations. Sleep medications may exhibit analogous pharmacokinetic characteristics to acetaminophen, where space-induced pharmacokinetic alterations delay drug absorption, necessitating dose escalation to maintain therapeutic sleep efficacy. However, such dose increases amplify their dose-dependent adverse effects—including impaired cognition, vigilance degradation, and operational performance decline—constituting an imperative consideration for astronaut health and mission success. It is important to note that these mechanistic insights into suboptimal efficacy remain hypothetical, as no direct studies have measured hypnotic pharmacokinetics in space. Furthermore, the absence of real-time medical supervision necessitates self-assessment-based medication decisions, while interindividual variability in ethnicity, genetic polymorphisms, space adaptation responses, and drug–drug interactions introduces confounding variables in medication metabolism studies, substantially elevating the complexity of extraterrestrial pharmacokinetic investigations.

For long-duration space missions, drug stability under space radiation is another concern. NASA’s Pharmaceutical Stability Initiative prioritizes lunar/Mars mission drug integrity studies [19]. An ISS study documented the effects of 550-day spaceflight exposure on the chemical potency of Zolpidem and Melatonin [20]. Following 550 days (≈18 months) of storage in ISS, Zolpidem retained 100.6% of its labeled active pharmaceutical ingredient (API) content, conforming to United States Pharmacopeia (USP) 2012 specifications (90–110%), with 0.29% total impurities (below the 0.5% threshold) and no abnormal degradation. These findings indicate that Zolpidem exhibited no significant chemical instability under ISS environmental parameters (20–26 °C, 30–50% relative humidity, and cumulative radiation exposure ≈ 70 mGy), demonstrating stability comparable to terrestrial storage conditions. In contrast, melatonin stored aboard ISS for the equivalent duration plus 11 months overage showed reduced API content (89.2%, below USP lower specification limit) with 10 degradation products totaling 0.96% impurities, including 5 individual impurities exceeding USP limits (>0.1%). Notably, the absence of ground control groups and multi-timepoint analyses precludes definitive exclusion of microgravity or chronic radiation effects. While current data suggest ISS environmental controls (thermal and humidity regulation) effectively maintain pharmaceutical stability for most compounds, future investigations must implement controlled experiments to validate long-term space environment impacts on drug integrity.

## 3. Light Therapy

As a non-pharmacological intervention, light therapy utilizes artificial illumination mimicking natural light to modulate circadian rhythms, effectively mitigating sleep disturbances and preserving operational performance for astronauts during space exploration.

Light therapy offers distinct advantages in space environments. Circadian rhythm disruption and sleep disorders can perturb hormonal homeostasis, thereby adversely affecting mental and behavioral functions through neuroendocrine dysregulation [21,22]. By directly regulating hormone levels, light therapy adjusts internal circadian timing, suppresses or delays sleep onset, and improves sleep quality with minimal interference from microgravity, radiation, or other space stressors. Light regulates biological rhythms through two pathways: classical photoreceptor pathways (rods and cones) and non-visual pathways mediated by intrinsically photosensitive retinal ganglion cells (ipRGCs) expressing melanopsin. This activation modulates the suprachiasmatic nucleus (SCN), the central circadian pacemaker, to control pineal melatonin production and aligning biological rhythms [23,24]. Light’s effects depend on wavelength and intensity: blue light (446–477 nm peak) maximizes melatonin suppression (with peak inhibition at 424 nm violet light) [25], while longer red wavelengths (631 nm) enhance melatonin secretion. Additionally, low-intensity light mildly suppresses melatonin, whereas high-intensity light causes significant but incomplete suppression. Intermittent exposure can also modulate melatonin secretion and shift circadian phases [26]. For example, 1000 lux light reduces melatonin to daytime levels, and even 350 lux significantly lowers nighttime melatonin. Notably, the sleep environment matters: the sleep quality can be affected by light exposure even with eyes closed, as 5 lux reduces rapid eye movement (REM) sleep duration and prolongs REM latency, with 10 lux amplifying these effects.

NASA first applied light therapy during the 1990 STS-35 Shuttle mission, where night crew members underwent a week of 10,000 lux bright white light exposure with simulated dawn–dusk transitions at Johnson Space Centre. This intervention successfully realigned the crew’s melatonin rhythms, demonstrating its ability to synchronize biological clocks with mission schedule [27]. Subsequently, the phototherapy protocol was systematically implemented across NASA’s Space Shuttle operations. In 2016, the ISS deployed the Solid-State Lighting Assemblies (SSLAs) system which enables controlled light exposure for circadian regulation through programmable spectral outputs. SSLA is a spectral LED consisting primarily of blue light with a 450–500 nm peak and red-yellow light with a 600–650 nm peak. A total of three modes have been designed for SSLA: General Mode balanced blue-red-yellow light for neutral melatonin inhibition; Bedtime Mode dominant red-yellow light to facilitate sleep by reducing blue light suppression; and Phase Shift Mode with high-intensity blue light with red-yellow enhancement to boost alertness and shift rhythms [28,29].

However, challenges persist. Firstly, individual variability necessitates personalized protocols. While SSLAs did not cause ocular strain in Treichel’s study [30], intensive phototherapy can induce transient side-effects like photophobia, headache, and vertigo [31,32]. That is, light therapy requires an individualized protocol, which is affected by many factors, such as the characteristics of the light, the duration of the treatment, the age of the subject, gender, genetics, etc. [31]. Currently, precision phototherapy integrating biometric analytics and adaptive algorithms represents a possible solution. Fares Siddiqui et al., funded by the National Space Biomedical Research Institute (NSBRI), worked on a sleep tracker based on non-contact technology. This contactless system quantifies sleep architecture through thoracic movement analysis and motion intensity monitoring, synchronizing data with mobile platforms to optimize daytime phototherapy parameters [33]. Astronauts could use such technology to tailor light exposure, during both day and night, to their unique sleep patterns and light responses. Secondly, light therapy provides temporary relief rather than curative effects. Studies by Campbell et al. show its benefits diminish after treatment stops, with melatonin levels recovering within 15 min of exposure cessation [34,35]. In order to cope with this problem, space lighting is shifting from reactive therapy to proactive prevention; the 2023 SpaceX Crew-7 mission tested dynamic circadian panels mimicking Earth’s diurnal and weather-dependent light cycles, aiming to stabilize intrinsic rhythms through programmable spectral sequences [36]. Thirdly, scheduling light exposure, typically in the morning or daytime, may conflict with mission demands, highlighting the need for integrated work–rest plans that prioritize both task execution and circadian alignment.

It is worth noting that the possibility exists for phototherapy to be used in combination with other therapeutic intervention. While no synergistic or antagonistic interactions between light therapy and other treatments (e.g., melatonin) have been reported in space study, their combined use shows additive effects in a terrestrial clinical investigation [37]. Further research into combinatorial therapies with sleep and rhythm regulators is warranted to maximize efficacy in space environments.

## 4. Scheduling Optimization

NASA recommends a daily in-orbit sleep duration of 8.5 h for astronauts, including 2 h of pre-sleep preparation and 1.5 h of post-sleep recovery. However, current astronaut sleep averages 6 h nightly, substantially below the thresholds recommended by the National Sleep Foundation and the American Academy of Sleep Medicine for maintaining satisfactory alertness, performance, and health. On the ISS, shift work to ensure 24 h system coverage often disrupts sleep patterns [38]. Studies demonstrate that optimizing work–rest schedules can increase sleep duration, improve sleep quality, and reduce chronic fatigue in space.

Napping serves as a spontaneous strategy for schedule optimization. ISS crew members frequently use daytime naps proactively to improve sleep quality [39]. While no astronaut-specific studies exist, terrestrial research on night-shift work (e.g., among nurses, physicians, and long-haul flight crews) indicates that properly timed naps can mitigate cognitive decline and sleepiness from shift work and sleep deprivation. Inappropriate napping, however, may disrupt circadian rhythms [40] and induce additional sleep inertia. These outcomes correlate with nap type (preventive vs. compensatory), timing, depth, duration (different nap opportunity durations), and individual differences:Nap Typology: Compensatory naps (taken after sleep deprivation) effectively reduce performance deficits, improving cognitive function and physical performance while alleviating perceived fatigue [41]. Proactive naps (pre-night shift) also enhance alertness, work quality, and workplace harmony [42,43];Timing: Early afternoon naps align best with human circadian rhythms, yielding superior results compared to other time slots [44]. Nighttime naps are less effective, possibly due to heightened sleep inertia sensitivity during nocturnal periods [45];Sleep Depth: The role of slow-wave sleep (SWS) in nap efficacy remains inconclusive [46]. Excessive SWS, however, may disrupt subsequent nighttime sleep duration and architecture [47], impairing circadian rhythms and sleep quality;Duration: Nap duration influences both intervention efficacy and sleep inertia severity. Studies demonstrate cognitive improvements from 10 min naps, while sleep inertia emerges in both 10 min nighttime naps and 30 min daytime naps [48,49];Individual Variability: Gender and age modulate nap effects. Females exhibit greater memory improvements from napping [50], while older adults face higher risks of circadian disruption with frequent napping [51].

In summary, short (30 min) shallow naps in the early afternoon before night shifts should be the preferred choice for astronauts to incorporate into their schedules. Light exposure, noise control, and temperature adjustments may alleviate sleep inertia, though the reliability of daytime napping as a scheduling tool requires further validation.

Terrestrial studies also highlight shift rotation patterns (clockwise and counterclockwise) as potential optimization strategies. While all shift work disrupts circadian rhythms, clockwise (CW) rotations reduce in-shift sleepiness, increase total sleep, and improve night-shift alertness compared to counterclockwise (CCW) rotations [52,53,54]. However, conflicting evidence suggests that, compare to CCW, CW rotations may cause greater negative effectsfor night shift workers without affecting sleep quality, underscoring the need for more research.

As human spaceflight missions extend into deep space exploration, communication delays pose significant challenges to traditional ground control-led scheduling models [55,56]. Astronauts must frequently adjust schedules under communication delays to adapt to evolving mission priorities, making enhanced crew autonomy critical. Future schedules will likely prioritize astronaut self-management to balance task-rest time, ensuring sufficient sleep while achieving mission objectives, especially for long-duration autonomous missions like deep space and Mars exploration. Therefore, decision-support tools are essential. The Fatigue Avoidance Scheduling Tool (FAST™), originally developed for aircrew scheduling optimization, could be adapted with mission-specific constraints to improve prediction accuracy and utility for astronaut self-scheduling. Biomathematical models to predict fatigue [57] could further assist in optimizing rest periods, particularly for long-duration autonomous missions to Mars and beyond. Studies indicate non-professional schedulers struggle with complex constraints [58], making such tools critical for maintaining performance under demanding conditions.

## 5. Human Flora Therapy

Existing circadian interventions for astronauts (pharmacotherapy, phototherapy, schedule adjustments) face limitations including side-effects and limited long-term efficacy. Emerging research highlights the gut microbiota’s critical role in sleep regulation through multiple biological pathways. The NASA twin study revealed that long-term space residency alters gut microbial composition and diversity [59], with subsequent studies confirming similar changes in space and space-analog environments [60,61,62,63,64]. Unlike exogenous chemical interventions, gut microbiota intervention represents an adjustment to the endogenous symbiotic homeostasis, making strategies to modulate microbial communities. This approach has emerged as a promising research direction for managing sleep disorders such as insomnia through systemic regulation of circadian and neuroendocrine functions.

The human microbiome comprises diverse symbiotic microorganisms including bacteria, archaea, viruses, and fungi. Different microorganisms form a rich microbial community in the human body, which mainly inhabits the gastrointestinal tract [65]. Gastrointestinal flora plays essential roles in external and internal barrier protection, homeostasis, and immunomodulation. Individual flora can be easily altered by different exogenous and endogenous factors, such as medications, diet, health status, hygiene, and microorganisms of the surrounding environment. Dysregulation of this microbial community structure can lead to local and systemic diseases [66], including sleep disturbances linked to central nervous system disorders. Direct evidence demonstrates bidirectional interactions: Voigt et al. found showed high-sugar, high-fat diet, and circadian disruption alter gut microbiota in mice [67], while Thaiss demonstrated that circadian disruption could be transferred via fecal microbiota transplantation (FMT), indicating microbial influence on host circadian rhythms [68]. In the same year, Leone et al. further showed gut microbiota modulates murine circadian clocks, germ-free mice exhibiting distinct hepatic and cerebral clock gene expression profiles compared to conventionally colonized counterparts [69]. Collectively, these findings establish bidirectional communication between host circadian rhythms and gut microbiota.

The microbiota–gut–brain axis (MGBA) is hypothesized as a key communication pathway, involving neurotransmitter modulation, immunoregulation, enteric nervous system activity, and short-chain fatty acids (SCFAs) [70]. This framework has translated into three therapeutic approaches for sleep disorders: FMT, psychobiotic administration, and dietary modification.

FMT, which transfers microbial communities from healthy donors to normalize recipient gut flora, is clinically used for drug-resistant infections and inflammatory conditions, such as inflammatory bowel disease (IBD), irritable bowel syndrome (IBS), and autoimmune disorders [71,72]. A 2023 study by Fang et al. conducted the first real-world study evaluating the implementation and efficacy of FMT for sleep disorders. Their findings demonstrated FMT reduced Insomnia Severity Index (ISI) from 17.31 to 5.38 and Pittsburgh Sleep Quality Index (PSQI) from 14.56 to 6.63 in chronic insomnia patients, with 76.47% achieving therapeutic endpoints [73]. This study provided preliminary evidence supporting the acceptability and feasibility of FMT as a novel intervention for sleep disorders, offering an alternative to traditional insomnia therapies. While promising, FMT may face challenges in addressing sleep disorders among astronauts, primarily due to operational impracticalities. In terrestrial settings, FMT is a relatively simple procedure with a short operational duration. In spaceflight, however, there are logistical challenges in transporting equipment and fresh donor samples, limited crew size for stable microbial sources, and potential adverse effects (e.g., constipation, diarrhea) that could impact high-stakes missions [72].

In 2013, Dinan and colleagues defined probiotics with potential therapeutic applications in mental health as a novel category of probiotics—“psychobiotics” [74]. Compared to conventional gut microbiota, psychobiotics demonstrate advantages in alleviating mental health issues, including sleep disorders such as insomnia. We have compiled research reports from the past decade exploring the effects of psychobiotics on sleep in Table 1. Although clinical applications of psychobiotics for treating insomnia remain underdeveloped, most existing studies report their positive therapeutic effects on sleep disturbances. Interestingly, the majority of literature indicates that psychobiotics improve sleep not by extending sleep duration, but rather by shortening sleep latency, enhancing sleep quality, and reducing daytime sleep dysfunction.

Though untested in astronauts, probiotic programs for immune and muscle health suggest feasibility [89,90]. Compared to traditional interventions, psychobiotics are safer, non-addictive, and associated with anxiety reduction, but space applications face challenges: 1. Uncertainty stability under microgravity/radiation (short-term ISS storage of *Lactobacillus casei* showed viability, but long-term data are lacking [91]; 2. formulation constraints, dairy-based carriers require refrigeration, while freeze-dried capsules or space-adapted diets need evaluation [92]; 3. individual variations in gut microbiota composition, dietary habits, and health status can lead to heterogeneous responses to psychobiotics, which posing challenges for standardized space protocols.

Dietary modification represents a non-invasive strategy, as food components and timing influence gut microbiota [93,94]. Clinical evidence has demonstrated successful applications of dietary interventions in reducing disease risks or pathological damage, highlighting their significant potential in preventive medicine. For addressing astronauts’ sleep disorders, dietary modulation holds unique advantages. Dietary modulation eliminates the need to transport additional medical equipment or pharmaceuticals into space, avoids risks associated with pharmacological side-effects or device malfunctions, and allows real-time adjustments based on mission requirements and individual responses. Nevertheless, challenges persist in space applications. The “black box” nature of food–microbiome interactions complicates mechanistic clarity, as demonstrated by studies showing divergent microbial responses to identical diets. Individual variations in gut microbiota composition (e.g., Prevotella-dominant vs. Bacteroides-dominant enterotypes) necessitate customized dietary plans, which poses logistical challenges in confined spacecraft environments. Additionally, nutrient stability under space radiation and microgravity remains unconfirmed, necessitating further research.

In summary, gut microbiota interventions offer innovative solutions for astronaut sleep disorders, but translation to space requires addressing environmental stability, individualization, and operational feasibility. Integrating these approaches with existing countermeasures may enhance circadian resilience for long-duration missions.

## 6. Traditional Chinese Medicine Treatment

In recent years, Traditional Chinese medicine (TCM) has emerged as a promising modality for addressing circadian rhythm disorders and sleep disturbances in astronauts, offering therapeutic potential to counteract space stress-induced emotional and cognitive impairments linked to sleep dysfunction [95,96]. For example, Huang et al. [97] demonstrated that *Gastrodia elata* (Tianma) reverses circadian disruption-induced learning and memory deficits in murine models by reducing serum malondialdehyde (MDA) levels and enhancing hippocampal superoxide dismutase (SOD) activity. The Space Nourishing Heart Pill formulation, a formulation containing *Crataegus*, *Panax ginseng*, *Acanthopanax gracilistylus*, etc., regulates the endocrine and nervous systems, effectively alleviating space adaptation syndrome, including insomnia, etc. [98]. Notably, *Panax ginseng*, a key ingredient, exhibits bidirectional regulatory properties, enhancing alertness while promoting sedation. Researchers have developed ginsenoside-based orally disintegrating tablets for circadian phase modulation [99], leveraging their rapid buccal absorption, ideal for aerospace environments where water access may be limited.

TCM offers distinct advantages for space sleep medicine through three core mechanisms. First, its holistic approach aligns with the complex multifactorial nature of astronaut sleep disorders [100]. Space missions expose crew members to microgravity confinement, and other stressors that disrupt multiple physiological systems [95]. TCM addresses these through formula customization and personalized therapy, enabling multisystem, multilevel, and multitarget regulation to improve sleep quality [100]. Second, TCM’s emphasis on safety and “medicinal-food homology” ensures low toxicity while maintaining efficacy [99]. Many TCM formulas can be integrated into dietary plans, offering nutritional value and palatability with minimal metabolic burden compared to conventional synthetic pharmaceuticals. This makes them suitable for long-term use in space, where systemic regulation of physiological functions is critical for sustained health. For instance, *Ginseng* and *Acanthopanax* species (used historically by Soviet cosmonauts as health supplements) contain sleep-promoting compounds that support circadian stability [101]. Third, TCM formulations may enhance the efficacy of Western medications while mitigating side-effects. Through their gentle pharmacological profiles, TCM interventions not only synergistically improve drug efficacy but also reduce dependency on potent Western medications. For instance, raw and parched *Ziziphus jujuba var. spinosa* seeds inhibit spontaneous locomotor activity in murine models and potentiate pentobarbital sodium-induced sedative–hypnotic responses, demonstrating synergistic anti-insomnia effects [102]. However, such interactions require caution, as co-administration or improper dosing may increase drug accumulation risks, potentially leading to adverse effects like excessive sedation.

While TCM holds promise for space applications, clinical translation requires systematic validation of its pharmacological profiles under space conditions and formulation optimization through controlled trials. Further research is needed to establish safety, efficacy, and compatibility with existing aerospace medical protocols, ensuring its integration as a complementary strategy for managing astronaut sleep health during long-duration missions.

## 7. Discussion

As human space exploration advances, an increasing number of astronauts are undertaking long-duration missions, during which circadian rhythm disruption and sleep disturbances are frequently reported among crew members. Inadequate sleep duration and quality pose risks to astronauts’ physical and mental health, as well as mission success. Current countermeasures for sleep disorders primarily include pharmacological interventions, light therapy, and work–rest schedule optimization [2,103]. However, light and schedule adjustments show limited efficacy, making sedative–hypnotic medications the most commonly used approach despite their challenges.

Current spaceflight pharmacotherapy relies largely on terrestrial drug administration principles [104], necessitating urgent research to clarify how space environments impact pharmacokinetics (PK) and pharmacodynamics (PD), for evidence-based extraterrestrial dosing guidelines. PK/PD studies typically use four approaches: human recumbent simulations, rodent hindlimb unloading (HLU) models, in-flight human studies, and animal spaceflight experiments. Head-down tilt (HDT) bedrest simulations of microgravity have produced inconsistent results, with some studies reporting altered drug PK parameters [105,106,107,108], and others finding no changes [109,110], highlighting HDT’s limitations in replicating real spaceflight PK/PD dynamics [111,112,113,114]. Validation studies comparing bedrest and actual spaceflight data remain lacking, undermining the translational value of ground-based models. Tail-suspended rodents mimic cephalad fluid shifts but show divergent hepatic cellular morphology and enzyme expression compared to short-term spaceflight-exposed animals [115,116], indicating HLU models inadequately reflect systemic space adaptations.

Given the limitations, in-orbit pharmacokinetic data from astronauts are now prioritized for scientifically rigor [117]. However, human space PK studies remain scarce. Sara Eyal [118] compiled five such studies, all relying on saliva-based drug concentration measurements, a method flawed by unpredictable drug diffusion and poor correlation with plasma concentrations [119,120]. Constraints include sample size, timing, drug classes, administration routes, and potential drug–drug interactions further limit insights. Future research should adopt alternative biomarkers or alternatives, such as electroencephalogram (EEG) for objective neuronal activity monitoring in the brain produced by internal or external stimuli [121] and wearable sensors for continuous physiological tracking [122,123], particularly for real-time monitoring of heart rate, physical activity, and sleep relevant to the assessment of sedative–hypnotic PK.

Cytochrome P450 (Cyp450) enzymes, critical for hepatic metabolism, and as mentioned above members of the CYP family, are involved in the metabolism of several drugs, including Zolpidem. Animal spaceflight studies indicate that microgravity exposure induces differential alterations in the expression and activity of CYP450 isoforms [18,108,124,125], but extrapolation to human responses is unvalidated. Human spaceflight data show minimal hepatic microsomal enzyme activity changes overall [125,126,127,128]. Physiologically based pharmacokinetic (PBPK) models offer a bridge between animal and human data, enabling predictions of drug behavior under altered physiology [129]. Zhang Yang’s team established PBPK models for rats under simulated weightlessness to extrapolate human PK for folic acid and Zolpidem tartrate [130]. To further advance research in space pharmacokinetics, Microphysiological systems (MPSs), also known as organ-on-a-chip systems, already deployed on the ISS, provide more physiologically relevant platforms for space PK research [131]. Enhancing international collaboration and data sharing will facilitate systematic, controlled in-space PK experiments to optimize dosing and develop astronaut-tailored therapeutics.

Drug stability is critical for long-duration missions (≥2 years). While ISS temperature/humidity are ideal for drug storage, space radiation may accelerate drug degradation [132]. Low-temperature storage conditions may have a protective effect on drugs during spaceflight, and this approach has been shown to be successful during radiation sterilization [133,134,135,136]. Some researchers have also proposed storing agents in a shielded compartment (a highly shielded compartment made of thick aluminum or other radiation-protective material on a probe vehicle) to reduce radiation exposure [134,137,138]. TCM, with its renewable herbal sources and adaptability, offers advantages. Herbal plants can provide continuous drug supplies, while TCM’s personalized, syndrome-specific approach addresses interindividual variability. For instance, the Chinese medicine Compound Space Nourishing Pill containing hawthorn, ginseng, prickly five-pronged ginseng, tangerine peel, and animal bone powder, can alleviate the discomfort caused by weightlessness. Ginseng and Acanthopanax can increase hemoglobin content, promote red blood cell production, and regulate endocrine/nervous systems and may enhance Western drug efficacy while reducing side-effects [98].

The search for low-side-effect sleep interventions is a priority in aerospace medicine. The current study reported that the strategies related to intestinal flora in ground experiments can play an important role in regulating circadian rhythms and improving sleep. Compared with drugs, the use of “psychotropic probiotics” for the treatment of sleep disorders has the advantages of being safer, well-tolerated, non-addictive, and reproducible, but the effectiveness of this approach for astronauts has not been reported. It is worth noting that, unlike sleep medications, psychotropic probiotics, as a class of biologically active agents, should take into special consideration the effects of the space environment on their activity. Space factors such as microgravity, radiation, and sub-magnetic fields have an important impact on the storage of psycho-probiotics in space transport, and studies have shown that probiotics appear to be more active in appropriate food matrices than in capsules. In addition, heat inactivation of the flora is a potential pathway to avoid the effects of the space environment. For FMT, the confined space and social environment is not conducive to its use in space. However, the human microflora is not limited to the gut, but includes the oral, respiratory, skin, urinary, and reproductive tracts. There are significant changes in the diversity and relative abundance of oral flora in the tongues of patients with chronic insomnia compared to healthy populations. As the second largest flora in the human body, oral flora may contribute to the research and development of sleep disorders, leading to the search for flora transplantation techniques that are more suitable for use by astronauts. Although numerous studies now show that the use of flora can improve sleep, human flora is a dynamic process of change, and how to appropriately use the right flora for intervention is still immature, and the mechanism process of remodeling the intestinal flora for dysbiosis needs to be studied in-depth.

A critical analysis of the current literature reveals substantial heterogeneity in diagnostic protocols for sleep disorders. Contemporary assessment methodologies predominantly encompass three modalities: (1) subjective evaluation (self-reported symptoms and sleep diaries), (2) psychometric instruments, and (3) objective physiological monitoring. The psychometric inventory includes, but is not limited to, the following validated scales: Athens Insomnia Scale (AIS), Beck Depression Inventory (BDI), General Health Questionnaire-28 (GHQ-28), Insomnia Severity Index (ISI), Piper Fatigue Scale (PFS), Pittsburgh Sleep Quality Index (PSQI), Richards–Campbell Sleep Questionnaire (RCSQ), Self-Rating Sleep Scale (SRSS), State–Trait Anxiety Inventory (STAI), Visual Analogue Scale (VAS), and the Verran Snyder–Halpern Sleep Scale (VSH). Objective measurements involve polysomnography (PSG), circadian rhythm biomarkers (melatonin secretion patterns, core body temperature oscillations), and actigraphic monitoring. This methodological diversity, compounded by the absence of standardized diagnostic criteria, has precluded consensus on categorical classification systems for sleep disorders. Of particular note, the European Sleep Research Society has established evidence-based guidelines for insomnia diagnosis and treatment (ESRS Consensus Document 2017, *J Sleep Res* [139]), which provide a potential framework for extrapolation to aerospace medicine contexts. Developing integrated assessment matrices, combining subjective, neurophysiological, and chronobiological data, will optimize therapeutic strategies and enable rigorous comparisons of countermeasures.

In conclusion, addressing astronaut sleep disorders requires integrating diverse countermeasures, prioritizing space-adaptive PK/PD research, and advancing diagnostics. Ground-based models and in-space trials must be complemented by international collaboration to ensure evidence-based, safe, and effective solutions for long-duration missions.

## Figures and Tables

**Table 1 ijms-26-05179-t001:** Research reports exploring the effects of psychobiotics on sleep.

Author (Year)	Population	Intervention (Strain, Dose, Route)	Study Design	Duration	Outcome Measures	Intestinal Microbiota Alterations	Sleep-Related Outcomes	Other Psychological Stress-Related Phenotypes
Monoi et al. (2015) [75]	Healthy adults (N = 68)	Sake Yeast Powder GSP6 (Compressed Tablets) 125 mg/tablet 4 tablets/day	Randomized, double-blind, placebo-controlled, crossover trial	4 days	EEG, OSA, GH secretion	Not reported	EEG: Increased Delta Power in First SWS Cycle; OSA:Improved subjective feeling of “sleepiness upon waking”; Activated human A_2a_ receptors; Upregulated GH secretion during sleep	- Not reported
Nakakita et al. (2016) [76]	Males aged 41–69 (N = 17)	*L. brevis* SBC8803 (capsule) 25 mg/day	Non-randomized, double-blind, placebo-controlled crossover pilot study	10 days	EEG, AIS, Sleep diary, BDI	Not reported	EEG: Increased Delta Power in below-average individuals, AIS and EEG: NS, Sleep diary: Reduced nocturnal awakenings/movement	- Not reported
Takada et al. (2017) [77]	Healthy fourth-year medical students under exam stress (N1 = 46, N2 = 48)	*L. casei Shirota* (Fermented Milk) 100 mL/day	Randomized, double-blind, placebo-controlled trial	8 + 3 Weeks *	EEG, OSA, Subjective Anxiety	Not reported	EEG: Increased N3 Percentage maintained, Increased Delta Power in first SWS cycle, OSA: Increased Score, Improved alleviation of sleep quality decline, sleep quality restoration	- Not reported
Nishida et al. (2017) [78]	Medical students participating in cadaver dissection course (N = 32)	*L. gasseri* CP2305 (Fermented Milk Beverage), 1 × 10^10^ cells/190g, 190g/day	Randomized placebo-controlled trial	5 weeks	GHQ-28, Zung-SDS, HADS STAI, PSQI, 100-mm VAS EAT 26, Saliva Testing, 16S rRNA pyrosequencing (V6-V8)	Decreased Bacteroides vulgatus; Increased Dorea longicatena	Increased PSQI latency/duration scores	Reduced male diarrhea-like symptoms
Marotta et al. (2019) [79]	Healthy adults Aged 18–35 (N = 38)	Probiotic mixture (*L. fermentum* LF16, *L. rhamnosus* LR06, *L. plantarum* LP01, and *B. longum* BL04, Maltodextrin), 4 × 10^9^ CFU/AFU, 2.5 g/packet, 1 packet/day	Randomized, double-blind, placebo-controlled trial	6 weeks	LEIDS-R, STAI, BDI-2, POMS, PSQI, TCI BIS/BAS, LOT-R	Not reported	PSQI: NS, subjective reports: Improved sleep quality	Increased LEIDS-R acceptance scores, Decreased POMS depression subscale scores
Moloney et al. (2021) [80]	Healthy males Aged 18–30 (N = 30)	*B. longum* AH1714 (Capsule), 1 × 10^9^ CFU/day	Randomized, placebo-controlled, repeated measures, crossover intervention study	8 weeks + 8 weeks ^#^	FFQ, IPAQ, GI-VAS, PSQI, PSS, BDI-II, Cognitive Performance CANTAB, PASA, CAR	α/β-diversity changes, species diversity, relative abundance: NS	Decreased PSQI Scores; Improved sleep duration/quality	Decreased PSS, BDI-II, PASA, Improved Neurocognitive performance post-acute stress
Lee et al. (2021) [81]	Healthy Adults with Subclinical Symptoms of Depression, Anxiety, and Insomnia (N = 156)	Probiotic NVP-1704 (*L. reuteri* NK33 and *B. adolescentis* NK98, Capsule) 2.5 × 10^9^ CFU/500mg (NK33: 2.0 × 10^9^ CFU, NK98: 0.5 × 10^9^ CFU), 2 capsules/dose, once daily	Randomized, double-blind, placebo-controlled trial	8 weeks	SRI, BDI-II, BAI PSQI, ISI, Stress Response Inventory, Blood Biomarker Testing, 16S rRNA pyrosequencing	Increased Bifidobacteriaceae/Lactobacillaceae; Decreased Enterobacteriaceae	Decresed PSQI/ISI scores; Improved sleep efficiency/latency	SRI, BDI-II, BAI: Decresed depression/anxiety scores
Ho et al. (2021) [82]	Participants Aged 20–40 with Self-reported Insomnia, (N = 40)	*L. plantarum* PS128 (capsule) 2 capsules/dose, once daily	Randomized, double-blind, placebo-controlled pilot trial	30 days	PSG, PSQI, ISI, ESS, BDI-II, BAI, STAI, MEQ, VAS	Not reported	PSQI, ISI, ESS Scores: NS, Improved deep sleep quality, Decresed total bed time and REM percentage, Decreased awakening frequency in N3, Increased N3 Percentage	Decresed BDI-II and BAI Scores
Zhu et al. (2023) [83]	Healthy Senior Students Under Exam Stress (N = 60)	*L. plantarum* JYLP-326 (Maltodextrin), 1.5 × 10^10^ CFU/packet, 1 packet/dose, twice daily	Randomized, double-blind, placebo-controlled trial	3 weeks	HAMA-14, AIS-8, HDRS-17 16S rRNA pyrosequencing	Improved restoration of gut dysbiosis and fecal metabolome disorder, changes in fecal metabolites: Decreased Ethyl Sulfate, Increased Cyclohexylamine	Decreased AIS-8 scores	Decreased HAMA-14, HDRS-17 Scores
Mäkelä et al. (2023) [84]	Healthy Participants Aged 18–40 Preparing for Semester Exams, (N = 190)	*L. paracasei* Lpc-37 (Capsule), 1.56 × 10^10^ CFU/day	Randomized, triple-blind, placebo-controlled trial	10 weeks	STAI, CAR, VAS PSS, BL-VAS, PSQI IPAQ, 16S rRNA pyrosequencing	Fecal microbiota diversity or composition: NS	PSQI: Decreased sleep duration scores, Decreased sleep disturbance scores	BL-VAS: reduced Alertness
Lan, Y. et al. (2023) [85]	Participants Diagnosed with Stress-induced Insomnia, (N = 40)	*B. breve* CCFM1025 (Maltodextrin), 10^9^ CFU/day	Randomized, double-blind, placebo-controlled trial	4 Weeks	PSQI, AIS, CAR ACTH, Serum Measurement	Not reported	Decreased PSQI Scores, Decreased AIS Scores, Improved sleep quality	- Not reported
Chan, H.H. et al. (2023) [86]	Participants with Sleep Disorders and Emotional Symptoms, (N = 68)	Novel E3 Probiotic (*L. acidophilus* GKA7, *L. casei* GKC1, *L. helveticus* GKS3, *L. plantarum* GKM3, B. GKB2, and *B. longum* GKL7, Capsule) 2 × 10^11^ CFU/day	Self-controlled before–after study, 8 weeks	8 Weeks	PSQI, GAD-7, PHQ-9 16S rRNA pyrosequencing	Increased relative abundance of Bifidobacterium, *Lactobacillus acidophilus*, *Lactobacillus helveticus*, and *Lactobacillus plantarum*	Decreased PSQI Scores, Improved subjective sleep quality	Decreased GAD-7, PHQ-9 Scores
Patterson, E. et al. (2024) [87]	Adults with Impaired Sleep Quality Aged 18–45, (N = 89)	*B. longum* 1714 (Capsule), 1 × 10^9^ CFU/day	Randomized, double-blind, placebo-controlled, parallel group, two-arm (allocation ratio 1:1) clinical trial	8 Weeks	PSQI, ESS, WASO, SF-36, PSS, HADS GASS	Not reported	PSQI: Decreased sleep quality scores and daytime dysfunction scores, ESS, Diary reports, actigraphy: NS	SF-36: Increased Energy/Vitality Trend
Mutoh, N. et al. (2024) [88]	Healthy Participants Aged 20–64, (N1 = 30, N2 = 30)	*Bifidobacterium breve* M-16V (Powdered Compressed Stick), 1 × 10^10^ CFU/stick, 2 sticks/day	Randomized, double-blind, placebo-controlled, parallel group clinical trial	6 Weeks	POMS 2, STAI, SDS, AIS, CFS, metabolite measurement 16S rRNA gene sequencing analysis	No significant differences observed between groups	Decreased AIS Scores, Improved sleep quality	Reduced LIR Values, Increased mean bowel movement frequency, Increased metabolites of the gut microbiota: pipecolic acid levels

ACTH: Adrenocorticotropic Hormone; AIS: Athens Insomnia Scale; BDI: Beck Depression Inventory; BIS/BAS: Behavioral Inhibition System/Behavioral Activation System; BL-VAS: Bond–Lader Visual Analog Scale; CAR: Cortisol Awakening Response; CANTAB: Cambridge Neuropsychological Test Automated Battery; CFS: Chalder Fatigue Scale; EEG: Electroencephalogram; EAT: Eating Attitudes Test; ESS: Epworth Sleepiness Scale; FFQ: Food Frequency Questionnaire; GAD: Generalized Anxiety Disorder; GASS: Glasgow Antipsychotic Side-effect Scale; GH: Growth Hormone; GHQ: General Health Questionnaire; GI-VAS: Gastrointestinal Visual Analogue Scale; HADS: Hospital Anxiety and Depression Scale; HAMA-14: Hamilton Anxiety Rating Scale-14; HDRS-17: Hamilton Depression Rating Scale-17; IPAQ: International Physical Activity Questionnaire; ISI: Insomnia Severity Index; LEIDS-R: Leiden Index of Depression Sensitivity-Revised; LIR: Lymphocyte Irradiation Response; LOT-R: Life Orientation Test-Revised; MEQ: Morningness–Eveningness Questionnaire; OSA: Obstructive Sleep Apnea; PASA: Post-Acute Stress Assessment; PHQ: Patient Health Questionnaire; POMS: Profile of Mood States; PSG: Polysomnography; PSQI: Pittsburgh Sleep Quality Index; PSS: Perceived Stress Scale; REM: Rapid Eye Movement; SDS: Self-Rating Depression Scale; SF-36: 36-Item Short Form Health Survey; SRI: Stress Response Inventory; STAI: State–Trait Anxiety Inventory; SWS: Slow-Wave Sleep; TCI: Temperament and Character Inventory; VAS: Visual Analogue Scale; WASO: Wake After Sleep Onset. * The study selects the test as a node because it is a controllable and repeatable event of acute psychological stress. The intervention plan for the first eight weeks and the final three weeks is the same; the difference lies in the focus of observation during the stress exposure and recovery stages. # This study utilized a crossover design, conducted in two 8-week intervention phases; Phase 1: Group 1 (G1) received placebo, Group 2 (G2) received probiotic; Phase 2: Groups received alternating treatments.

## Data Availability

The authors declare that no data were generated and submitted and hence not applicable.

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
