# Peer review of "Circadian Disruption and Sleep Disorders in Astronauts: A Review of Multi-Disciplinary Interventions for Long-Duration Space Missions"

_ijms, 2025, doi:10.3390/ijms26115179_

Round 1
Reviewer 1 Report
Comments and Suggestions for Authors
This is an interesting review paper on sleep disturbances in astronauts with focus on therapeutic modalities. Manuscript itself is well written with scientific merit. Topic is of interest for wide reader of the journal.
Some issues to be ameliorated:
- in introduction, it is stated that this is systematic review, but according to the rest of the manuscript it is a narrative review (quite good one). Therefore, this statement should be changed. Otherwise, full methodology for systematic reviews etc according to PRISMA guidelines.
- circadian disruption could impact hormonal status resulting in anxiety, what could be observed in astronauts in extraterrestrial environment (e.g.PMID: 31269081) . It should be emphasized alongside the light therapy option
- minor language errors should be ameliorated (e.g. "Light therapy has its own advantages in the space environment. Light therapy has its own advantages and disadvantages in the space environment.")
Reviewer 2 Report
Comments and Suggestions for Authors
Astronauts on deep space missions often experience disrupted circadian rhythms and sleep deficits due to microgravity, radiation, and work-related stress. These disruptions can impair cognitive function and reduce operational performance. Current countermeasures include pharmacological treatments, light therapy, and optimized work schedules. Emerging strategies like gut microbiota modulation and Traditional Chinese Medicine also show potential for improving sleep quality during long-duration spaceflight. This review evaluates the key strengths and weaknesses of these approaches, aiming to provide guidance for future studies. The review is well written and I don’t have any major issue with it. I just have a few minor suggestions.
Please correct the English. There are several typos like the following:
- Change Prevalent to prevalence in line 39.
- Line 50 – start a new sentence with meanwhile.
- Change evidence to evidences – line 82.
- Line 102 – Has anyone tried cocktails of GABA agonists and inhibitors for glutamatergic signaling?
- There is repetition – line 196.
- The whole paragraph starting at line 196 can be shifted in the beginning of this section.
- Extra full stop on line 243.
- Please add references at 343?
- What are the scores in the last column of the table? Either explain them or simplify them please.
Need to be improved.
Round 2
Reviewer 1 Report
Comments and Suggestions for Authors
My comments have been well-addressed in the revised manuscritp.